# Performance Prediction and Optimization of Nanofluid-Based PV/T Using Numerical Simulation and Response Surface Methodology

**DOI:** 10.3390/nano14090774

**Published:** 2024-04-28

**Authors:** Sreehari Sreekumar, Supriya Chakrabarti, Neil Hewitt, Jayanta Deb Mondol, Nikhilkumar Shah

**Affiliations:** 1Centre for Sustainable Technologies (CST), Belfast School of Architecture and the Built Environment, Ulster University, Belfast BT15 1ED, Northern Ireland, UK; nj.hewitt@ulster.ac.uk (N.H.); jd.mondol@ulster.ac.uk (J.D.M.); n.shah@ulster.ac.uk (N.S.); 2Nanotechnology and Integrated Bio-Engineering Centre (NIBEC), Ulster University, Belfast BT15 1ED, Northern Ireland, UK; s.chakrabarti@ulster.ac.uk

**Keywords:** nanofluid, CFD, response surface method, ANOVA, optimization

## Abstract

A numerical investigation was carried out in ANSYS Fluent^®^ on a photovoltaic/thermal (PV/T) system with MXene/water nanofluid as heat transfer fluid (HTF). The interaction of different operating parameters (nanofluid mass fraction, mass flow rate, inlet temperature and incident radiation) on the output response of the system (thermal efficiency, electrical efficiency, thermal exergy efficiency, and electrical exergy efficiency) was studied using a predictive model generated using response surface methodology (RSM). The analysis of variance (ANOVA) method was used to evaluate the significance of input parameters affecting the energy and exergy efficiencies of the nanofluid-based PV/T system. The nanofluid mass flow rate was discovered to be having an impact on the thermal efficiency of the system. Electrical efficiency, thermal exergy efficiency, and electrical exergy efficiency were found to be greatly influenced by incident solar radiation. The percentage contribution of each factor on the output response was calculated. Input variables were optimized using the desirability function to maximize energy and exergy efficiency. The developed statistical model generated an optimum value for the mass flow rate (71.84 kgh^−1^), the mass fraction (0.2 wt%), incident radiation (581 Wm^−2^), and inlet temperature (20 °C). The highest overall energy and exergy efficiency predicted by the model were 81.67% and 18.6%, respectively.

## 1. Introduction

Developing a variety of solar energy conversion systems, namely photovoltaic [1], photothermal [2], photochemical [3], photobiological [4] and other hybrid energy conversion [5] systems, is a growing area of research since solar energy is inexpensive, environmentally friendly, and secure. Photovoltaics have a huge potential to decarbonise future power production. However, the PV cell efficiency decreases by 0.4–0.5% with each degree rise in temperature [6]. Unlike conventional PV modules, and side-by-side PV/T, a hybrid PV/T system aids in the production of high-grade electricity and low-grade heat from the same module area. A typical hybrid PV/T system includes a PV module, an absorber sheet (fixed to the rear end of the PV module) and a circulating heat transfer fluid. The incident solar radiation on the top PV surface gets transmitted to the HTF by both conduction and convection modes of heat transfer [7].

Conventional liquid heat transfer fluids include deionized water, glycol (ethylene glycol, propylene glycol) [8], thermal oils [9], mixtures [8,10], PCM slurry [11,12], and ionic liquids [7,13]. Numerous experimental and numerical studies have evaluated the performance of hybrid PV/T with conventional fluids. Li et al. [14] performed an experimental and numerical investigation on the performance of a novel water-based PV/T system. Comparing the system to traditional PV/T, an improvement in electrical efficiency of roughly 2% was achieved. Lower heat conductivity and a lack of application-specific property tunability are the primary drawbacks of conventional HTFs. Research for developing efficient working fluids has contributed to the creation of microfluids [15] and nanofluids [16] that were developed by dispersing micro and nanoparticles in conventional HTFs. Studies show that integrating nanofluids with higher thermophysical characteristics in solar collectors could enhance the heat transfer performance of the system [8,17]. The overall efficiency enhancement of PV/T with conventional nanofluids synthesized by dispersing metal/metal oxide, carbon, and carbide/nitride nanomaterials was reported by various studies [8,18]. Recently developed MXene nanofluids were reported to be exhibiting considerable enhancement in thermal conductivity and heat transfer efficiency [19,20]. Sreekumar et al. [20] conducted thermo-optical characterization on MXene, carbon quantum dot (C-dot), and novel MXene/C-dot nanofluids. The study shows that thermal conductivity enhancement of around 50, 42.2, and 33.2% was achieved with MXene, hybrid and C-dot over base fluid. Few studies were reported on the application of MXene nanofluid in PV/Ts. Even though the cost involved with MXene-based nanofluid synthesis is comparable to other nanofluids (MWCNT or graphene) of similar thermal property enhancement, very little research has been reported on the adoption of MXene nanofluids in PV/Ts. A comprehensive experimental analysis of MXene-based PV/T system involves high experimentation time, cost, and nanomaterial wastage. An efficient and validated numerical model can overcome the limitations of an experimental study. Numerical studies on MXene-based PV/T systems are lesser. A PV/T collector using a sheet-and-tube configuration with MXene nanofluid was numerically simulated by Samylingam et al. [21]. When compared to aluminium oxide nanofluid, the study found improvements in thermal efficiency and a heat transfer coefficient of roughly 16% and 9%, respectively.

Very few studies were reported on the optimization of PV/T systems design and operating conditions. The most widely adopted methods for determining the effect of input variables on output responses include the design of experiments (DoE) [22,23] and artificial neural network (ANN) [24,25]. DoE is observed to provide the results in the minimum number of experiments [26]. DoE can also be extended to complex numerical models to predict output based on a minimum number of numerical data, thereby reducing the computation cost and time. Kazemian et al. [27] analysed a water-based PV/T system in series combination with a flat plate collector. The operating parameters of the system were optimized using the Taguchi grey relation-based technique. Optimization was purely based on the thermal energy efficiency achieved by the system. Very few reports were published on optimizing the system-operating parameters based on the exergy efficiency of nanofluid-based PV/T [28]. Khani et al. [29] conducted a multi-objective optimization of the design parameters of a water-based PV/T collector to improve electrical and thermal efficiency. A numerical model was simulated using finite volume analysis (FVM) and the data were used for optimization using a Genetic algorithm. The model is reported to enhance the thermal and electrical yield by 11.5% and 8.6%, respectively, in comparison to the existing optimised model [30]. Kazemian et al. [31] developed a CFD model of a PV/T-PCM system in ANSYS Fluent, and a predictive model was generated using RSM in Design Expert software. The study found the optimum PCM thickness (15 mm), melting temperature (298 K), ambient temperature (313 K) and solar irradiation (901 Wm^−2^) for the system. Hosseinzadeh et al. [32] employed the Taguchi method to evaluate the optimal operating parameters at which the ZnO/water-based PV/T system generates maximum electrical and thermal power. Coolant inlet temperature was observed to be the most influencing parameter as about a 16% enhancement in thermal efficiency of PV/T was reported with a reduction in inlet temperature by 20 °C.

Based on the critical analysis of the recent studies discussed above, MXene nanoparticle dispersions could significantly enhance the thermal properties of the base fluid. Moreover, investigation on MXene/water nanofluid as heat transfer fluid in PV/T systems is very rare. Also, a comprehensive experimental study on factors affecting the efficiency of a nanofluid-based PV/T system is costly and time-consuming. Hence, the study proposes a new optimization method by indirect coupling between ANSYS Fluent^®^ and Design Expert^®^ 11 software. This study numerically investigates the thermal and electrical characteristic performance of a hybrid PV/T collector with MXene/Water nanofluid. The numerical study was conducted under different operating conditions as defined by the Design Expert software. RSM and desirability functions were adopted for parametric study and optimization. Different factors (nanofluid mass flow rate, concentration, inlet temperature and solar radiation) affecting the efficiency of the system were studied numerically and the results were used for optimization. A numerical study on MXene/water-based PV/T system followed by RSM-based optimization has not been conducted to date.

## 2. Model Description and Methodology

### 2.1. Methodology and Geometric Model

The methodology adopted for performing the combined numerical simulation and optimization of the MXene nanofluid-based PV/T system can be observed in Figure 1. As seen from the figure, two main steps (numerical simulation and multi-objective optimization) are involved. In the first step, a numerical FVM analysis was performed on the studied computational domain. The geometric model consists of a sheet-and-tube absorber-based PV/T system, as visualized in Figure 2. MXene/Water nanofluid was selected as the heat transfer fluid. A single tube was considered for simulation to reduce the computational complexity [27]. The geometry was modelled in SpaceClaim and the components are in the order as shown in Table 1. The component dimensions, and material properties, as adopted from the literature, are also provided in Table 1 [27] (shown in Section 2.4). Grid independence test and validation were conducted on the model (as shown in Section 2.7). The validated numerical model was then simulated at various operating conditions to generate the required amount of data for optimization. The numerical data were then analysed using energy and exergy efficiency correlations for a PV/T system (as shown in Section 2.5). This data were used in the optimization process (Step 2), as shown in Figure 1. In this step, suitable optimization models were selected (explained in Section 3.1). The appropriate input parameters, output responses, and optimization objectives were selected for the optimization study (detailed in Section 3.2, Section 3.3, Section 3.4, Section 3.5).

### 2.2. Assumptions

The present numerical simulation was performed based on the following assumptions:➢Laminar, uniform, and steady flow was considered for the model;➢Nanofluid was assumed to be in a single-phase model to reduce the computational time and complexity;➢Thermal contact resistance between the PV/T layers was assumed to be negligible;➢The initial temperature of the system was assumed to be the same as the ambient temperature;➢Radiation and convection heat loss occurred from the top glass surface to the surroundings;➢Adiabatic condition was considered for the side and lowermost system surfaces;➢The PV layer absorbed all incoming solar energy received on the glass top surface as the glass cover and EVA were considered to be completely transparent;➢No-slip boundary condition was used at the fluid–absorber tube interface.

The details of the numerical model used in the study were explained in the previous work [33]. Reynolds number (Re) decreased with an increase in fluid viscosity. The highest Re of about 555, obtained for the least viscous fluid (water), fell within the laminar regime (<2300). The Re of nanofluids varied from 67–390 [33]. Hence, the flow was assumed to be laminar for nanofluids and base fluid. Constant thermophysical properties were taken for the PV/T module layers as their size was considerably less and their effect on the results was negligible [33,34].

### 2.3. Governing Equations

The conservation equations that govern the simulation are provided below in Equations (1)–(4) [33]. The information on fluid and flow parameters was obtained by solving the conservation and momentum equations. The data on heat transmission and component temperatures in the system were obtained by solving the energy equation. For this study, a second-order upwind scheme was selected to perform the interpolation as a higher-order accuracy would be obtained. 

Continuity equation
(1)$$\frac{𝜕\rho_{nf}}{𝜕t}+\nabla .(\rho_{nf}{\vec{V}}_{nf})=0$$

Momentum equation
(2)$$\frac{𝜕(\rho_{nf}{\vec{V}}_{nf})}{𝜕t}+\nabla .(\rho_{nf}{\vec{V}}_{nf}{\vec{V}}_{nf})=-\nabla P+\mu_{nf}(\nabla^{2}{\vec{V}}_{nf})+\rho_{nf}g+S$$

Energy equation for fluid
(3)$$\frac{𝜕(\rho_{nf}C_{p,nf}{{\vec{T}}_{nf}\vec{V}}_{nf})}{𝜕t}=\nabla .(k_{nf}\nabla {\vec{T}}_{nf})$$

Energy equation for solid component
(4)$$k_{s}\nabla^{2}\left(T_{s}\right)=0$$

### 2.4. Thermophysical Properties of Nanofluid

Empirical relations, as provided in Table 2, were used for predicting the effective density, thermal conductivity, specific heat, and viscosity of nanofluids at different concentrations. These equations were used to find the effective properties of nanofluids based on the base fluid properties and nanoparticle concentration. The major drawback faced during nanofluid modelling was the degradation of thermal conductivity with time due to the decrease in the stability of the colloidal solution. MXene nanofluid was selected as the heat transfer fluid and exhibited considerably higher colloidal stability and lower viscosity compared to other nanofluids [35].

In the above equations, $\rho_{nf}$, $C_{p,\;nf}$, $\mu_{\;nf}$, and $K_{nf}$ represent the density, specific heat capacity, viscosity, and thermal conductivity of a nanofluid. While $\varphi_{np}$, $C_{p,np}$, and $K_{np}$ indicate the nanomaterial’s concentration, specific heat and thermal conductivity values were used for calculating the nanofluid properties. The density, thermal conductivity and viscosity of the base fluid that were used in the model are represented by $\rho_{bf}$, $K_{bf}$, and $\mu_{bf}$, respectively. The thermophysical properties of the various concentrations of the MXene/water nanofluid are visualized in Figure 3.

### 2.5. Energy Efficiency Analysis

#### 2.5.1. Thermal Efficiency

The thermal efficiency of the nanofluid-based PV/T was evaluated using Equation (5), as provided below. The known values of nanofluid flow rate, inlet temperature, specific heat, incident radiation, and collector area were used. The thermal energy generated in the fluid due to the solar irradiation by heat transfer was calculated using Equation (6), after obtaining the nanofluid outlet temperature from the simulation. The aperture area of the PV/T, used for calculation, is provided in Table 3.
(5)$$\eta_{th}=\frac{Q_{u}}{A\times I}$$
(6)$$Q_{u}=m_{f}\times C_{p,f}(T_{out}-T_{in})$$

#### 2.5.2. Electrical Efficiency

The electrical efficiency of the PV/T system varied with the PV surface temperature and was calculated using Equation (7). The PV reference efficiency (18%), reference surface temperature (298 K) at standard test conditions, and cell temperature coefficient (0.0045) values, as shown in Table 3, were used to calculate the electrical efficiency variation with PV surface temperature [31,33]. Numerical simulation provided the value of cell temperature at each point during the operational time.
(7)$$\eta_{PV}=\eta_{ref}\times (1-\beta (T_{cell}-T_{ref}))$$

### 2.6. Exergy Efficiency Analysis

#### 2.6.1. Thermal Exergy Efficiency

Thermal exergy efficiency was calculated to find the exergy destruction associated with the system. Input exergy entering the domain is the exergy from solar radiation due to nanofluid inflow. Exergy leaving the system is contributed by heat loss from the PV/T surface to the surroundings and exergy lost during nanofluid outflow. The exergy balance calculations for the PV/T were performed as shown in Equation (8). Equation (9) was used to calculate the exergy received from incident radiation. The sun’s temperature used in the calculation was 5774 K, as shown in Table 3. The net thermal exergy gain by the fluid was calculated using Equation (10). By substituting Equations (9) and (10) in Equation (11), the thermal exergy efficiency of the PV/T system was calculated as follows:(8)$$E_{ex,in}=E_{ex,\;out}+E_{ex,\;loss}$$
(9)$$E_{ex,sol}=I_{sol}A\times (1-\frac{T_{amb}}{T_{sun}})$$
(10)$$E_{ex,\;th}=Q_{u}-\dot{m}C_{p}T_{amb}ln(\frac{T_{out}}{T_{in}})$$
(11)$$\eta_{ex,\;th}=\frac{E_{ex,\;th}}{E_{ex,sol}}=\frac{Q_{u}-\dot{m}C_{p}T_{amb}ln(\frac{T_{out}}{T_{in}})}{I_{sol}A\times (1-\frac{T_{amb}}{T_{sun}})}$$

#### 2.6.2. Electrical Exergy Efficiency

The electrical exergy of the PV/T system is the high-grade electrical energy output generated by the PV/T module [42]. The electrical exergy is calculated by dividing the electrical exergy of the system by incident exergy, as shown in Equation (13). The parameter values used in the calculation is provided in Table 3. Pumping power was not included in the equation as the effect of pumping power on the electrical efficiency, while using nanofluid was calculated and was found to be considerably small in comparison to the base fluid [33].
(12)$$E_{ex,ele}=IA\times \eta_{PV}$$
(13)$$\eta_{ex,\;ele}=\frac{E_{ex,\;el}}{E_{ex,sol}}=\frac{IA\times \eta_{PV}}{I_{sol}A\times (1-\frac{T_{amb}}{T_{sun}})}$$

### 2.7. Grid Independence Test and Validation 

Poly-hexacore shape with mosaic meshing technique was adopted for meshing the domain of the study, as shown in Figure 4. The element size adopted for the meshing was in the range of 0.0015 to 0.005 m. The mesh was refined for each case with a different number of elements. The number of elements of the domain varied from 1.5 to 5 × 10^5^. For each case, steady-state simulation was performed on the system under the same input parameter values for solar radiation, flow rate, and inlet temperature (470 Wm^−2^, 0.00136 kgs^−1^, and 305 K, respectively) were used for the analysis. The convergence criterion selected for continuity and momentum equations was 10^−5^, and for energy equation, the limit was set as 10^−6^. Negligible variation in outlet temperature and outlet velocity was observed for each case. As observed from Figure 5, only a 0.7 °C change in the outlet temperature was observed during grid-sensitive analysis. Also, a minute variation in outlet temperature of around 0.5% was noticed while changing the number of elements from 3.7 to 4.9 × 10^5^. For validating the present model, the experimental data from Selmi et al. [43] and numerical results from Khanjari et al. [34] were used. The inlet temperature (32–46 °C), solar radiation (470–542 Wm^−2^) and mass flow rate (0.00136 kgs^−1^) values from the study were adopted as operating conditions for simulating the proposed PV/T model. The dimensions of the 3D PV/T model were almost the same as those of the literature. On error analysis, it was observed that the relative variation in the results with those of experimental and numerical studies was only 8.97% and 2.75% respectively.

## 3. Results and Discussion

### 3.1. Response Surface Methodology

A typical optimization methodology faces the drawback of inefficiency in finding the interaction between multiple input parameters and output responses. Optimization using RSM helps to develop a statistical model for accurately predicting the responses of the model from the desired input parameters [44,45]. RSM-based optimization could be performed using the Design Expert software package. A mathematical model, as shown in Equation (14), would be used to define the relation between the studied input parameters and output responses.
(14)$$y=c_{0}+\sum_{i=1}^{n}c_{i}x_{i}+\sum_{i=1}^{n}c_{ii}x_{i}^{2}+\sum_{i=1}^{n}\sum_{j=1}^{n}c_{ij}x_{i}x_{j}+e$$

In the above equation, y, $c_{0}$, x, n, and e represent the output response, constant coefficient, design variable, number of parameters, and the model residual, respectively. Meanwhile, $c_{i}$, and $c_{ii}$, represent the linear and quadratic regression coefficients of the *i*^th^ factor, respectively. And the term $c_{ij}$ indicates the interaction of the *i*^th^ and *j*^th^ factors. The total number of data sets required for the analysis was calculated using Equation (15).
(15)$$n=2^{k}+2k+n_{cd}$$

In the above equation, *n*, *k*, and $n_{cd}$, represent the number of runs, input parameters and central design points used in the analysis. The steps adopted in the optimization process are visualized in Figure 1. Four input parameters (nanofluid concentration, flow rate, solar irradiation, and inlet temperature) were used for the optimization. Three levels of values were used for each input parameter to generate the data set for the analysis. The range of values used for each input variable is shown in Table 4. The analysis began with developing a design matrix for generating data for analysis. A design matrix with 27 sets of input parameter combinations was created by the software, as shown in Table 5. The numerical simulation of the nanofluid-based PV/T system was performed for all the combinations of operating parameters. According to the literature, the maximum concentration of MXene nanofluid that produces better thermophysical property improvement with the least amount of viscosity enhancement was chosen as the upper limit (0.2 wt%) [33]. The lower limit was selected to achieve at least a 5% average enhancement in thermal conductivity over a temperature range of 20–60 °C. Four output parameters/responses (thermal efficiency, electrical efficiency, thermal exergy efficiency and electrical exergy efficiency) were adopted for the multi-objective optimization. Based on the results from the simulation, the energy and exergy efficiencies were calculated and provided as output responses to the Design Expert software. Three data sets were of the same operating parameters (0.105 wt%, 60 kgh^−1^, 600 Wm^−2^, and 27.5 °C) to evaluate the data set repeatability. The central composite design (CCD) model of RSM was adopted for the present optimization as the design produces better accurate prediction models in a reduced number of data sets [26]. A desirability function is a mathematical tool in Design Expert that is coupled with RSM to perform the optimization of input values [46]. The range of desirability function varies from 0 (low value) to 1 (high value). A statistical model with a desirability value close to 1 is considered to have achieved the desired objective.

### 3.2. ANOVA Analysis

ANOVA analysis was performed to evaluate the significance of the statistical predictive model. The F-value and *p*-value are the two important criteria determining the model validity and the effect of interaction between factors on the output response [47]. The model is significant when its F-value is below 0.05. The *p*-value indicates the significance of the parameters used in the model. The higher the *p*-value, the greater the effect of that parameter on the response. The model fitness is predicted by three parameters: R-squared ($R^{2}$), adjusted R-Squared ($R_{adj}^{2}$), and predicted R-Squared ($R_{pred}^{2}$). The difference between $R_{adj}^{2}\;$ and $R_{pred}^{2}$ values should be less than 0.2 for the model to be valid. The total sum of the squares ($S_{tot}$) of the measured response (*y*), the sum of the squares of the model residuals ($S_{res}$), and the sum of the squares of the prediction error ($S_{pred}$) was calculated using Equations (16)–(18), respectively [47,48]. These parameters were used in Equations (19)–(21) to calculate the $R^{2}$, $R_{adj}^{2}$, and $R_{pred}^{2}$, respectively [47].
(16)$$S_{tot}=\sum_{i=1}^{n}{\left(y_{i}-\bar{y}\right)}^{2}$$
(17)$$S_{res}=\sum_{i=1}^{n}{\left(y_{i}-{\hat{y}}_{i}\right)}^{2}$$
(18)$$S_{pred}=\sum_{i=1}^{n}{\left(\frac{e_{i}}{1-h_{ii}}\right)}^{2}$$
(19)$$R^{2}=1-\frac{S_{res}}{S_{tot}}$$
(20)$$R_{adj}^{2}=1-\frac{S_{res}/(n-p)}{S_{tot}/(n-1)}$$
(21)$$R_{pred}^{2}=1-\frac{S_{pred}}{S_{tot}}$$

In the above equation, *y*, $\bar{y},$ and ${\hat{y}}_{i}$ represent the measured response, mean response value, and the predicted response values of the model. The residual ($e_{i}$) is the difference between the measured and the predicted response. The term $h_{ii}$ represents the *i*th diagonal element of the hat matrix H = $X{\left(X^{\prime }X\right)}^{-1}X^{\prime }$ [31,47,48].

#### 3.2.1. ANOVA Analysis of Thermal Energy Efficiency

The detailed ANOVA analysis data of thermal energy efficiency is shown in Table 6. The thermal energy efficiency model was found to be significant (as *p*-value < 0.0001). Independent variables, namely, a mass fraction, mass flow rate, solar radiation, and inlet temperature (represented by A, B, C, and D, respectively) and their respective dependent terms were used in the analysis. The significant model terms for thermal energy efficiency were A, B, D, CD, B^2^, and C^2^. The mass flow rate (F-value of 96.01) was found to be the significant factor that determines the thermal efficiency of the PV/T system. Afterwards, the parameters—the inlet temperature, mass fraction, and solar radiation—were associated with F-values of 36, 12.86, and 2.16, respectively. Equation (5) shows the influence of mass flow rate on the thermal energy efficiency of PV/T. As the data were generated numerically, the repeatability of the output response was ensured, and the model fit perfectly. Also, the difference between adjusted-R^2^ and predicted-R^2^ being less than the desired value of 0.2 ensured the model was valid. As the adequate precision value for the model (15.32) was significantly above the minimum required value of 4, the model could be used to predict the responses efficiently. The basic empirical model for thermal energy efficiency prediction was derived (as shown in Equation (22)). The predicted and actual data plot is shown in Figure 6. The values predicted using the equation are shown by the line. The numerical data which were used as input to the model are represented by coloured dots. As observed, the actual data and predicted data agreed with negligible variation, and they could be used for prediction.

#### 3.2.2. ANOVA Analysis of Electrical Energy Efficiency

The ANOVA of the electrical energy efficiency of nanofluid-based PV/T is detailed in Table 7 below. The model is significant (as *p*-value < 0.001) and valid (difference between R^2^ terms < 0.2). As inferred from the table, solar radiation (F-value of 6048.10) and inlet temperature (F-value of 1112.97) were the two independent variables that impacted the electrical efficiency of the PV/T. Analysis shows that even though a mass fraction has a significant effect on thermal efficiency, the electrical efficiency of the system is least influenced by a nanofluid mass fraction. The model developed for predicting the electrical efficiency of the MXene/nanofluid-based PV/T system is shown in Equation (22). The predicted values and the actual value are plotted in Figure 7, and it is observed to be coinciding along the 45° line with minimal variation.

#### 3.2.3. ANOVA Analysis of Thermal Exergy Efficiency

The ANOVA analysis is tabulated in Table 8 and the model is observed to be significant. A, B, C, D, BC, BD, and CD were the terms that had more influence on the output values. Among the input variables, solar radiation intensity (F-value of 591.71) had the highest impact on thermal exergy efficiency. The relationship of the variable on exergy efficiency could be observed from Equation (9). The next significant parameter that could affect the exergy efficiency was the mass flow rate of the nanofluid (F-value of 84.09). As observed from Figure 8, the predicted model data and actual data are in good agreement.

#### 3.2.4. ANOVA Analysis of Electrical Exergy Efficiency

The ANOVA analysis results of a modified model are catalogued in Table 9. Electrical energy and electrical exergy efficiencies have a similar trend in terms of influencing parameters. Solar radiation intensity (F-value of 6048.10) and inlet temperatures (F-value of 1112.97) were found to be the most dominating input variables affecting the electrical exergy efficiency of the PV/T system. Among the independent parameters, the mass fraction was observed to have the least importance. As observed, A, B, C, D, AB, and AC were the most significant parameters. The model was observed to be a good fit and predicted accurately, as the actual numerical data points and predicted values from the model were almost coinciding, as seen in Figure 9.

### 3.3. Model Prediction

The generated model is based on regression equations for predicting the output responses corresponding to the variables used. Four investigated responses ($\eta_{th}$, $\eta_{ele}$, $\eta_{ex,\;th}$, and $\eta_{ex,\;ele}$) were predicted using correlations created during the RSM-based ANOVA analysis. In this method of analysis using RSM, a second-order polynomial equation was used for model creation. Independent input variables ($\varphi ,\dot{m},\;I,\;\mathrm{and}\;T_{i})$ and the combination terms of the variables ($\varphi .\dot{m},\;\varphi .I,\;\dot{m}.I,\;\dot{m}.T_{i},\;\varphi .T_{i},\;I.T_{i},\;\varphi^{2},\;{\dot{m}}^{2},\;I^{2},\;\mathrm{and}\;{T_{i}}^{2})$ were used to coin the four equations. The general equation for all four output responses developed using the RSM analysis is given in Equation (22). The coefficients of the terms used in each equation are provided in Table 10 below.
(22)$$\mathrm{Response}=a_{0}+a_{1}\varphi +a_{2}\dot{m}+a_{3}I+a_{4}T_{i}+a_{5}\varphi .\dot{m}+a_{6}\varphi .I+a_{7}\dot{m}.I+\;a_{8}\;\dot{m}.T_{i}+a_{9}\varphi .T_{i}+a_{10}I.T_{i}+{a_{11}\varphi }^{2}+a_{12}{\dot{m}}^{2}+a_{13}I^{2}+a_{14}{T_{i}}^{2}$$

### 3.4. Impact of Input Parameters on Output Responses

#### 3.4.1. Interaction of Input Parameters on Thermal Energy Efficiency

The variation of thermal efficiency with each parameter is depicted in the 3D plot in Figure 10a–f. The 3D graphs represent the interaction of (a) $\varphi \;\mathrm{and}\;\dot{m},$ (b) φ and I, (c) $\dot{m}\;\mathrm{and}\;I,$ (d) $\dot{m}\;\mathrm{and}\;T_{i},$ (e) $\varphi \;\mathrm{and}\;T_{i},$ and (f) $I\;\mathrm{and}\;T_{i}$ on thermal energy efficiency. As inferred from the ANOVA analysis, the nanoparticle mass flow rate and nanofluid concentration have the highest dominance in the thermal efficiency of the PV/T system. For each 3D graph, two parameters were varied while keeping the base values of other input parameters constant. The central level values of the mass flow rate (60 kgh^−1^), mass fraction (0.105 wt%), solar radiation (600 Wm^−2^), and inlet temperature (27.5 °C) were used for all the 3D plots. As observed in Figure 10a,d,e, the thermal efficiency of the PV/T system increased considerably with the nanofluid mass flow rate. Also, the thermal energy efficiency was found to increase with the nanofluid mass fraction, as observed in Figure 10a–c. The centre point values of the input parameters are highlighted in the 3D contour plot, as observed in Figure 10a–f.

#### 3.4.2. Interaction of Input Parameters on Electrical Energy Efficiency

The interaction of parameters ($\varphi ,\;\dot{m},\;I,\;\mathrm{and}\;T_{i})$ on the electrical energy efficiency is shown in Figure 11a–f. Two input variables were altered to generate the 3D plots while keeping the other parameters constant. The same central level parameters, as explained in Section 3.1, were used. As observed in Figure 11b,d,f, the electrical efficiency of the MXene-based PV/T system was impacted significantly by incident solar radiation. The solar radiation intensity had a negative interaction, as the electrical efficiency decreased steeply with an increase in solar radiation intensity. This could be inferred from Equation (17) which was used for calculating the electrical efficiency of PV/T. Higher solar radiation intensity resulted in higher surface temperature and, consequently, lower electrical efficiency. The interaction effect of nanofluid inlet temperature on efficiency is shown in Figure 11c,e,f. The fluid inlet temperature is significant as it is the second most influencing parameter as observed from the ANOVA analysis. The graphs show that efficiency was observed to decrease with an increase in the inlet temperature. It is inferred from Figure 11f that the highest electrical efficiency would be attained when the nanofluid-based PV/T operated with the lowest inlet temperature and minimum solar radiation. Figure 11b shows that efficiency is almost constant with varying nanofluid mass fractions.

#### 3.4.3. Interaction of Input Parameters on Thermal Exergy Efficiency

The influence of input parameters on the thermal exergy efficiency of the system is shown in a 3D plot, as visualized in Figure 12. Empirical correlation connecting the nanofluid flow rate, inlet temperature, and incident solar radiation is provided in Equation (16). Solar radiation forms the major exergy inflow to the system domain. Exergy due to fluid flow also contributes to the input exergy of the system. Thermal exergy was found to increase with an increase in incident radiation (as observed in Figure 12b,d,f) and with a decrease in mass flow rate (as observed in Figure 12a,d,e). The highest exergy efficiency (above 2.2%) was observed in Figure 12d at maximum irradiation (100 Wm^−2^), least mass flow rate (30 kgh^−1^), and other parameters at base conditions. Observation from Figure 12a–c shows that the nanofluid mass fraction has the least impact on the thermal exergy efficiency of the system compared to other parameters.

#### 3.4.4. Interaction of Input Parameters on Electrical Exergy Efficiency

Six contour plots depicting the interaction of various parameters on the electrical exergy efficiency of the system are shown in Figure 13. A similar trend as that of electrical efficiency was observed with the interaction of parameters on electrical exergy efficiency. As inferred from Figure 13b,d,f, a small increase in incident radiation was found to decrease the exergy efficiency drastically. The nanofluid inlet temperature was also found to be negatively affecting the electrical exergy efficiency of PV/T. With the nanofluid mass flow rate and mass fraction being the least influencing factors on electrical exergy efficiency, the 3D plane in Figure 13a was almost a straight horizontal plane showing negligible variation.

ANOVA analysis data were used to calculate the percentage contribution (PC) of each input parameter on the output response and were calculated using Equation (23) [27,49]. The percentage contribution of each factor was calculated by dividing the sum of the squares of the factor (${SS}_{i}$) by the total sum of the squares of all factors used in the model.
(23)$$PC\;(\%)=\frac{{SS}_{i}}{\sum_{i=1}^{n}{SS}_{i}}$$

The percentage contribution of each independent factor and the combined contribution from all other dependent factors are shown in Figure 14. As observed from Figure 14a, the mass flow rate showed higher dominance of about 57.1% on the thermal energy efficiency of the system, followed by inlet temperature with 21.4%. As inferred from Figure 14b–d, the incident solar radiation was observed to have a profound effect on all other responses with a percentage contribution above 76%. All the factors exhibited a similar percentage of influence on electrical energy and exergy efficiencies, where solar radiation and inlet temperature had the most major effect. In the case of thermal exergy efficiency, mass flow rate was found to be the second most influencing factor with a contribution of 10.8%.

### 3.5. Optimization

Multi-objective optimization is complex, as the most significant factor for each response variable is different. The thermal energy efficiency of the PV/T has the highest dependence on mass flow rate, while its influence on electrical efficiency is minimal. The thermal exergy efficiency of the system was found to be largely influenced by solar irradiance, followed by mass flow rate. The electrical energy and exergy efficiency are dependent on solar irradiation. However, the effect of solar radiation on thermal efficiency is much less. The Design Expert software employs a desirability objective function to assess the optimization process. The range of desirability function values extends from 0 to 1. Zero is the least desirable and 1 is the required output response. However, the objective is to optimize and not to attain a desirability value of 1. The overall objective desirability function is given in Equation (24).
(24)$$D={{(d}_{1}.d_{2}\ldots .d_{n})}^{\frac{1}{n}}$$

In the above equation, $d_{i}$ refers to the individual desirability of responses. The total number of responses is represented by n. Conditions are assigned to the input parameter values to be in the selected range during optimization. Thermal and electrical energy and exergy efficiencies are assigned equal weights (3) and the same objective (maximizing the values). Three solution sets of optimum values of input parameters and corresponding output responses are catalogued in Table 11. The desirability graph is visualized as shown in Figure 15. The desirability of each case is also provided in the table. The results show that the PV/T could perform best with the nanofluid having a mass fraction and a flow rate of 0.2 wt% and 71.84 kgh^−1^, respectively, at an inlet temperature of 20 °C and with 581 Wm^−2^ incident radiation. At this operating condition, the system generated thermal and electrical energy efficiencies of about 65% and 16.6%, respectively. Thermal and electrical exergy efficiencies of the system were reported to be around 1% and 17.5%, respectively. The overall energy and exergy efficiency achieved by the MXene nanofluid-based PV/T system were 81.7% and 18.6%, respectively. The desirability index of the solution was reported to be 0.606. Hence, the optimization procedure was confirmed to be valid. Other than the above-mentioned optimal solution, a few other solutions were also generated. Three of them are tabulated in Table 11. The proposed RSM-based optimization technique can be used along with numerical simulation to predict the thermo-electrical performance of solar PV/T systems while reducing computational time and cost.

## 4. Conclusions

The numerical model of the MXene/water-based PV/T system was generated using ANSYS Fluent^®^. Fluent meshing was performed on the 3D model and validated with experimental and numerical data in the literature with a minimal error of around 2.7% and 8.9%, respectively. The influence of operating parameters on the PV/T system performance was studied using Design Expert software and a multi-objective optimization was performed using desirability function. The effect of nanofluid flow rate, concentration, incident radiation, and fluid inlet temperature on the thermal, electrical and heat transfer performance of the system was studied using the CCD model in RSM. ANOVA analysis was conducted to study the effect of various parameters on the energy and exergy efficiency of the system. Major findings from the study are explained below:
➢Analysis shows that thermal efficiency was significantly influenced by the nanofluid mass flow rate, as indicated by the corresponding highest F-value of 1073.31; incident radiation intensity with an F-value of 6048.10 was the significant factor affecting the electrical energy efficiency of the system;➢The thermal exergy efficiency of the PV/T was affected mainly by variations in solar irradiation; also, incident irradiation was the dominating parameter that determined the electrical energy and exergy efficiencies of the nanofluid-based PV/T system;➢The mass flow rate was found to have the highest percentage contribution of 57.1% on thermal energy efficiency, while solar radiation had the highest contribution, of above 76%, on thermal exergy, electrical energy and exergy efficiencies;➢The developed statistical model generated an optimum value for the mass flow rate (71.848 kgh^−1^), mass fraction (0.2 wt%), incident radiation (581.21 Wm^−2^), and inlet temperature (20 °C) at a desirability value of 0.611;➢The highest thermal and electrical energy efficiency that could be achieved at optimum conditions were around 65.03% and 16.67%, respectively. Also, the thermal and electrical exergy efficiency of MXene-based PV/T were observed to be about 1.01% and 17.58%.

Hence, the study shows the significance of MXene nanofluids in enhancing the thermal and electrical performance of the system. This research has a wider research scope for future direction. Future research would focus on incorporating the thermophysical properties of nanofluids that are obtained from experimental characterization. Research would be focussed on extending this simulation model and optimization approach on a large-scale PV/T system and their experimental validation.

## Figures and Tables

**Figure 1 nanomaterials-14-00774-f001:**
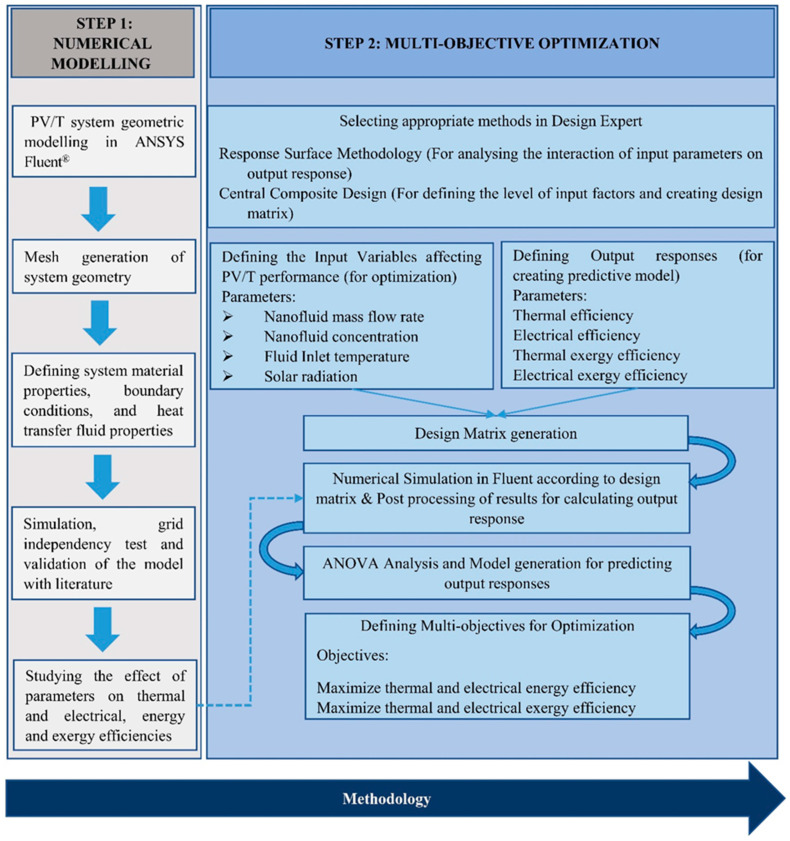
Numerical simulation and optimization methodology.

**Figure 2 nanomaterials-14-00774-f002:**
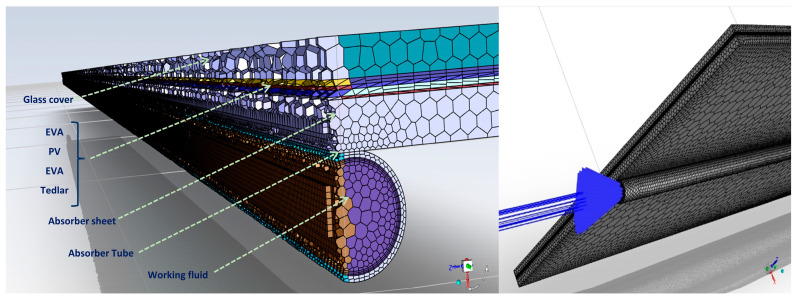
Geometric model developed in SpaceClaim for the present study.

**Figure 3 nanomaterials-14-00774-f003:**
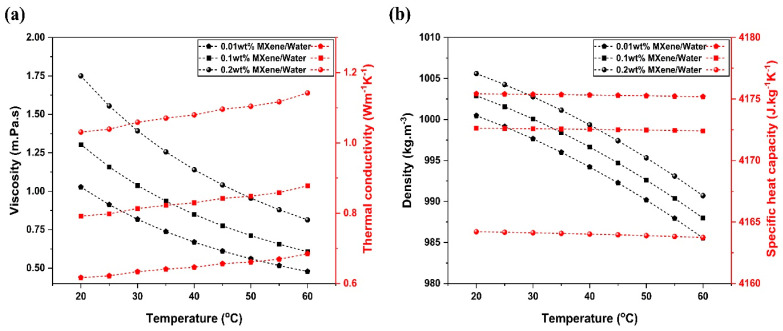
Temperature-dependent variation in (**a**) viscosity and thermal conductivity, and (**b**) the density and specific heat capacity of various concentrations of MXene nanofluid.

**Figure 4 nanomaterials-14-00774-f004:**
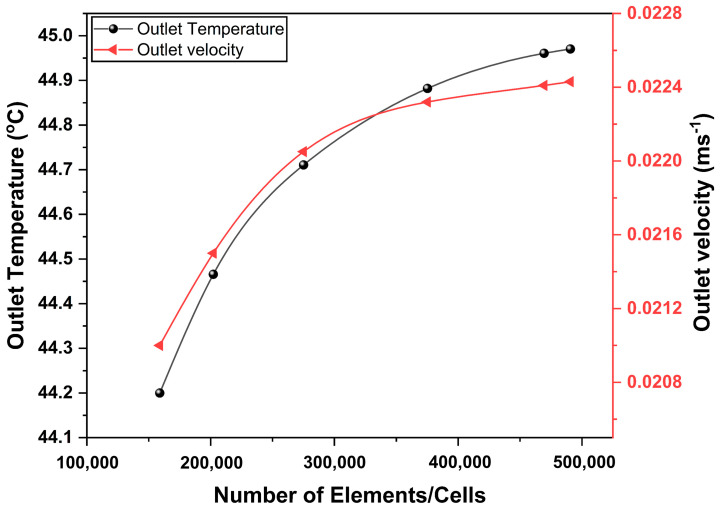
Grid independence test performed on the model at base conditions.

**Figure 5 nanomaterials-14-00774-f005:**
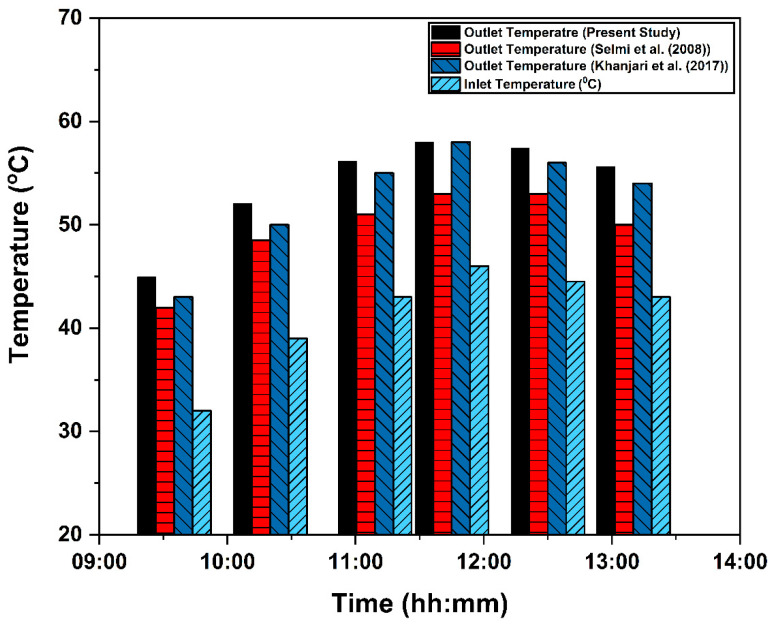
Validating the model with Selmi et al. [43] and Khanjari et al. [34].

**Figure 6 nanomaterials-14-00774-f006:**
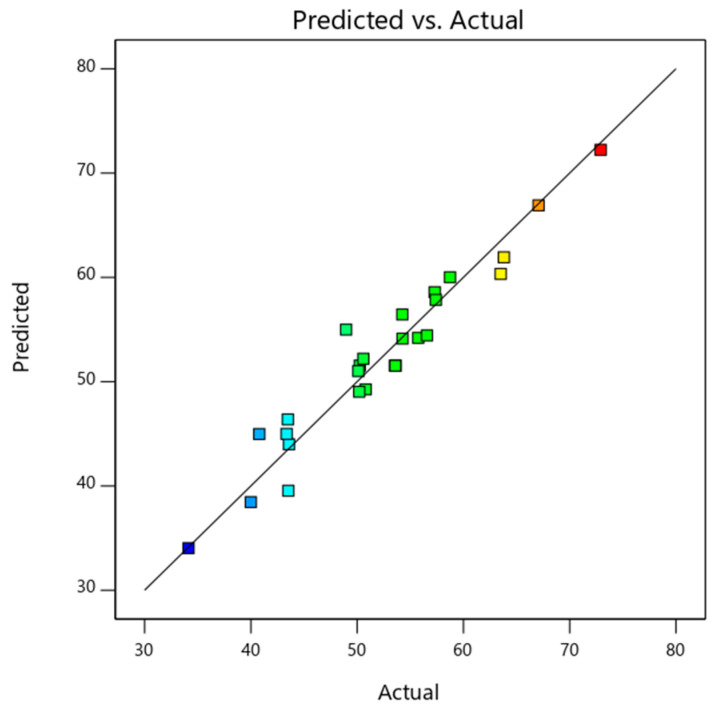
Predicted and actual data plot of PV/T thermal efficiency.

**Figure 7 nanomaterials-14-00774-f007:**
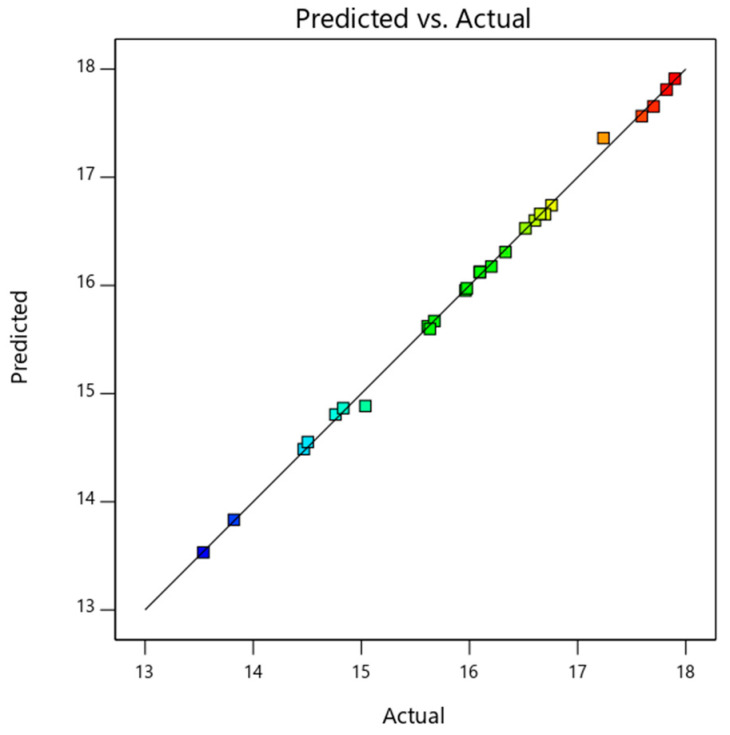
Predicted and actual data plot of PV/T electrical efficiency.

**Figure 8 nanomaterials-14-00774-f008:**
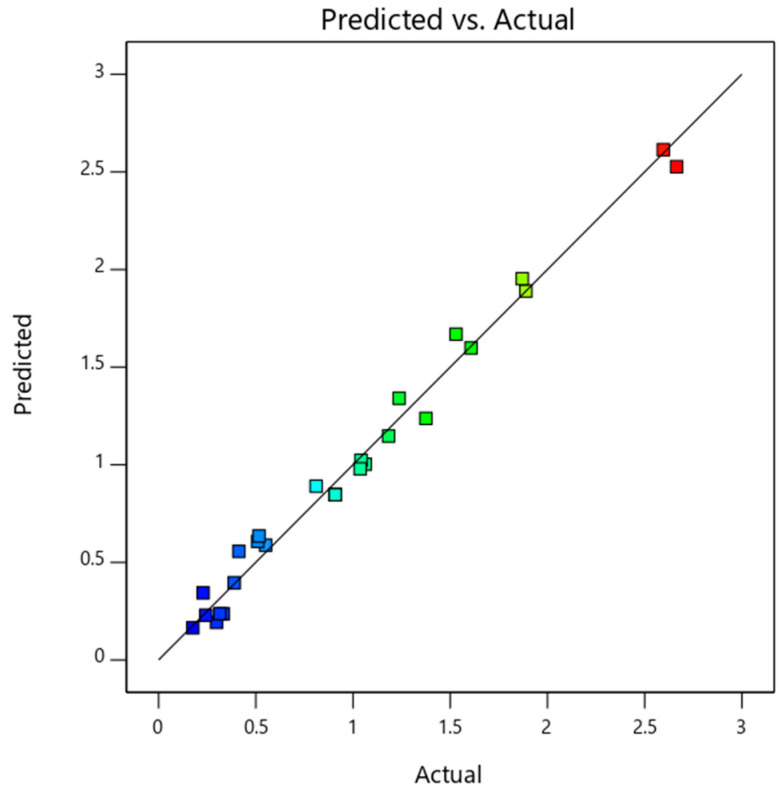
Predicted and actual data plot of PV/T thermal exergy efficiency.

**Figure 9 nanomaterials-14-00774-f009:**
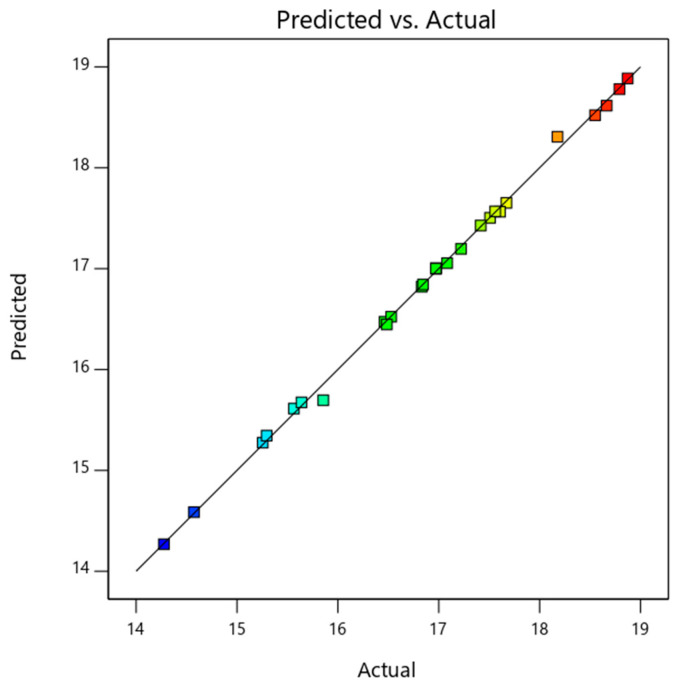
Predicted and actual data plot of PV/T electrical exergy efficiency.

**Figure 10 nanomaterials-14-00774-f010:**
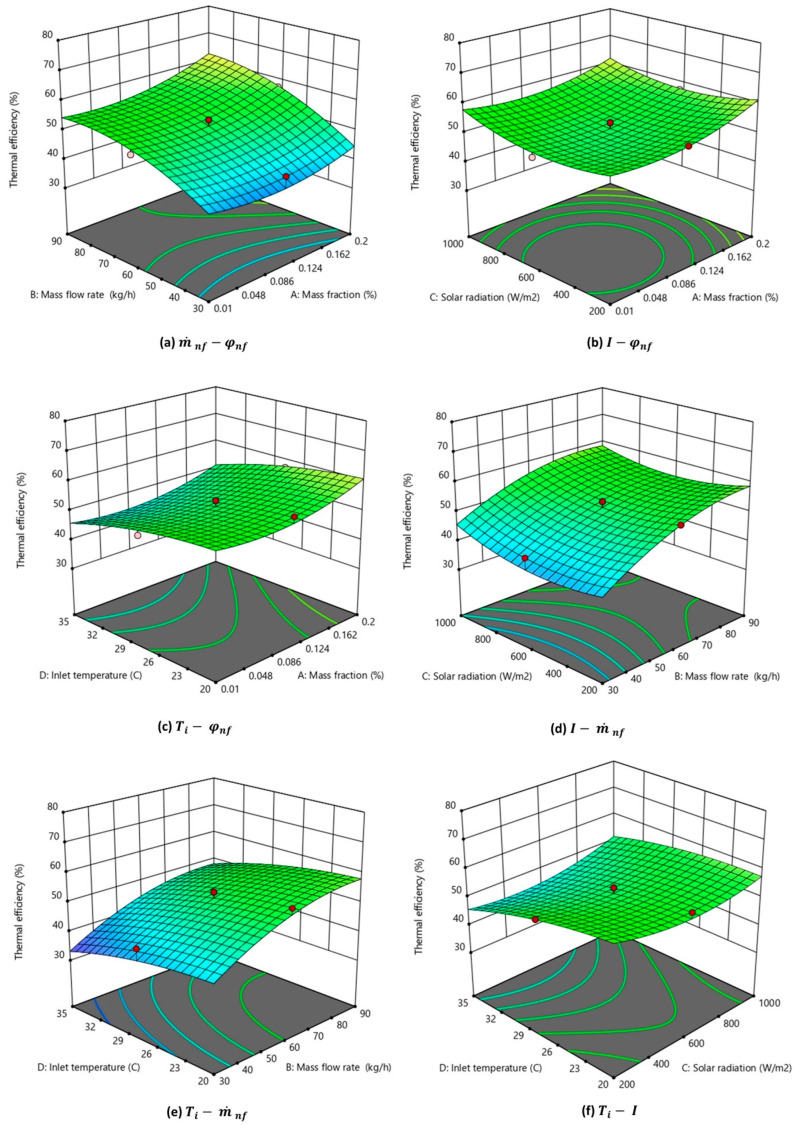
Three-dimensional graph depicting the interaction effect of multiple parameters on the thermal efficiency of the nanofluid-based PV/T system.

**Figure 11 nanomaterials-14-00774-f011:**
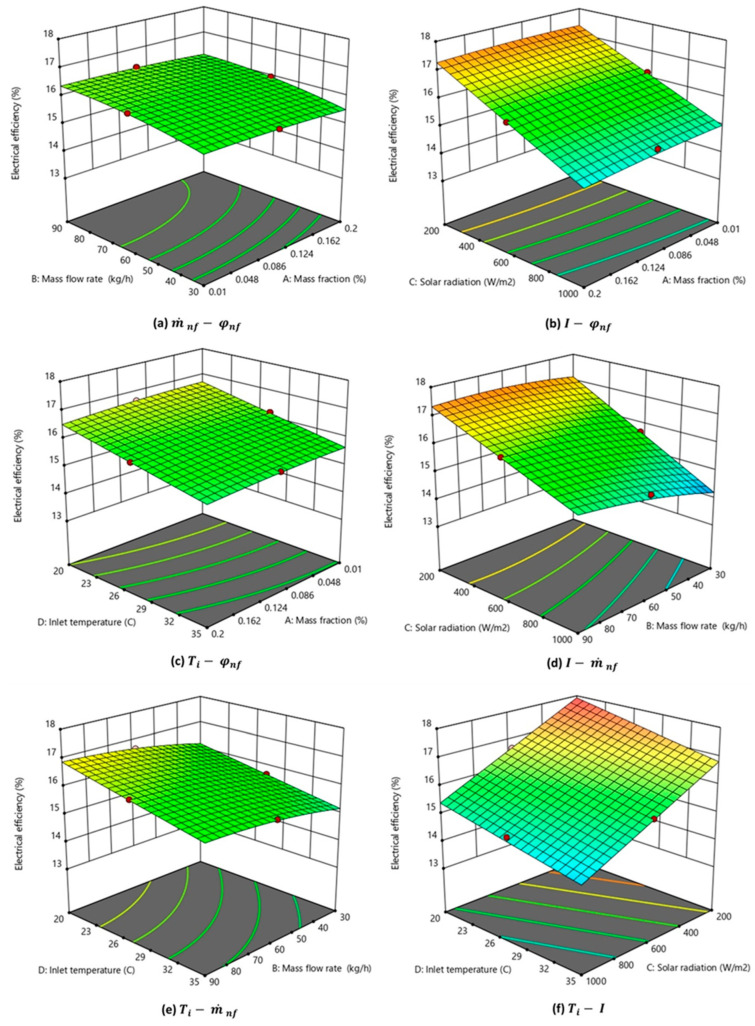
Three-dimensional graph depicting the interaction effect of parameters on the electrical energy efficiency of the nanofluid-based PV/T.

**Figure 12 nanomaterials-14-00774-f012:**
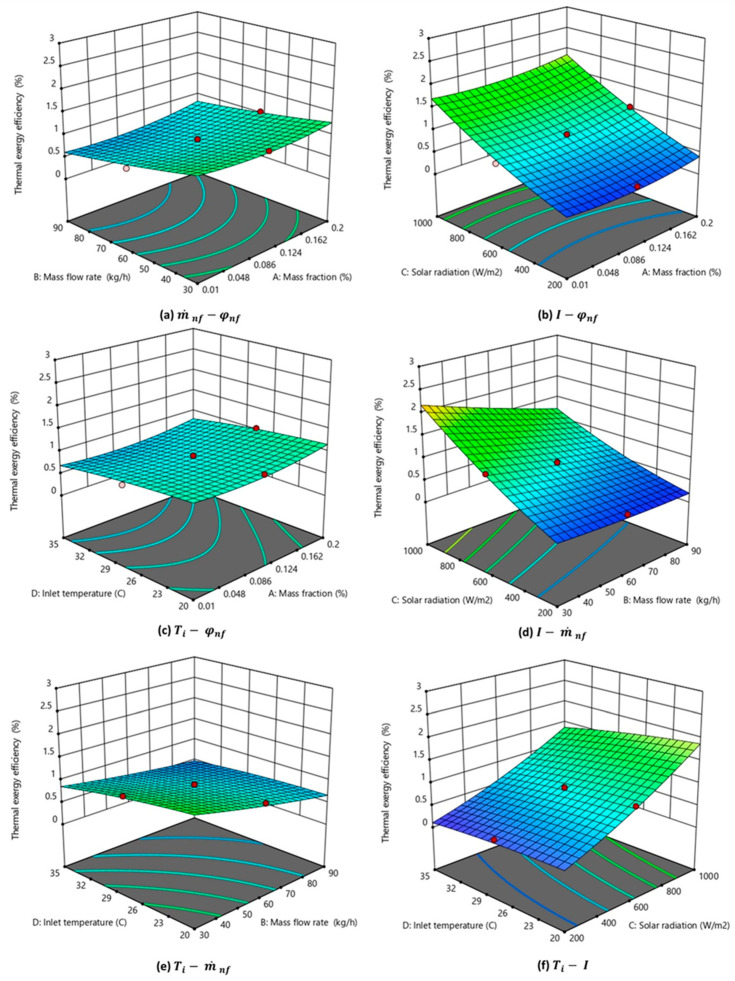
Three-dimensional graph depicting the interaction effect of parameters on the thermal exergy efficiency of the nanofluid-based PV/T.

**Figure 13 nanomaterials-14-00774-f013:**
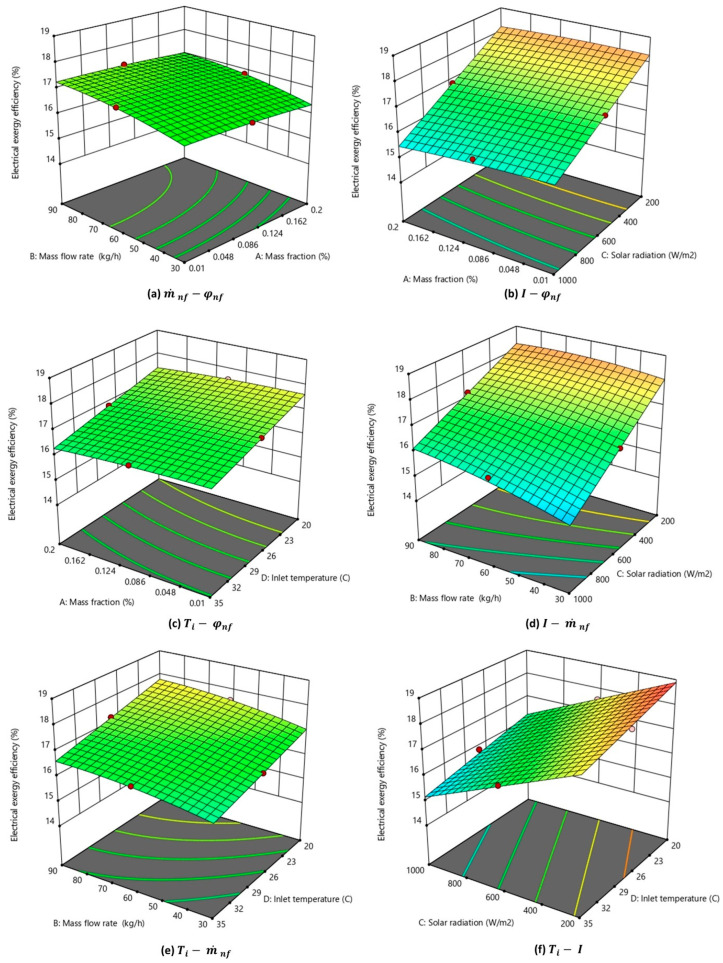
Three-dimensional graph depicting the interaction effect of parameters on electrical exergy efficiency.

**Figure 14 nanomaterials-14-00774-f014:**
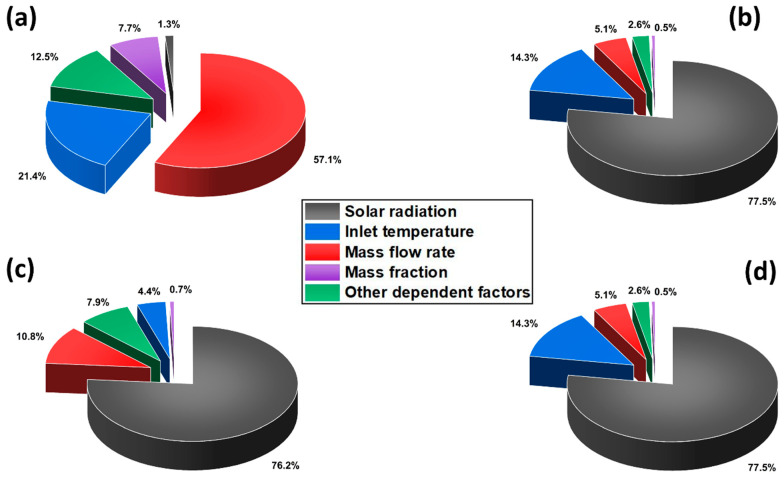
Percentage contribution of input factors on (**a**) thermal energy, (**b**) electrical energy, (**c**) thermal exergy, and (**d**) electrical exergy efficiencies.

**Figure 15 nanomaterials-14-00774-f015:**
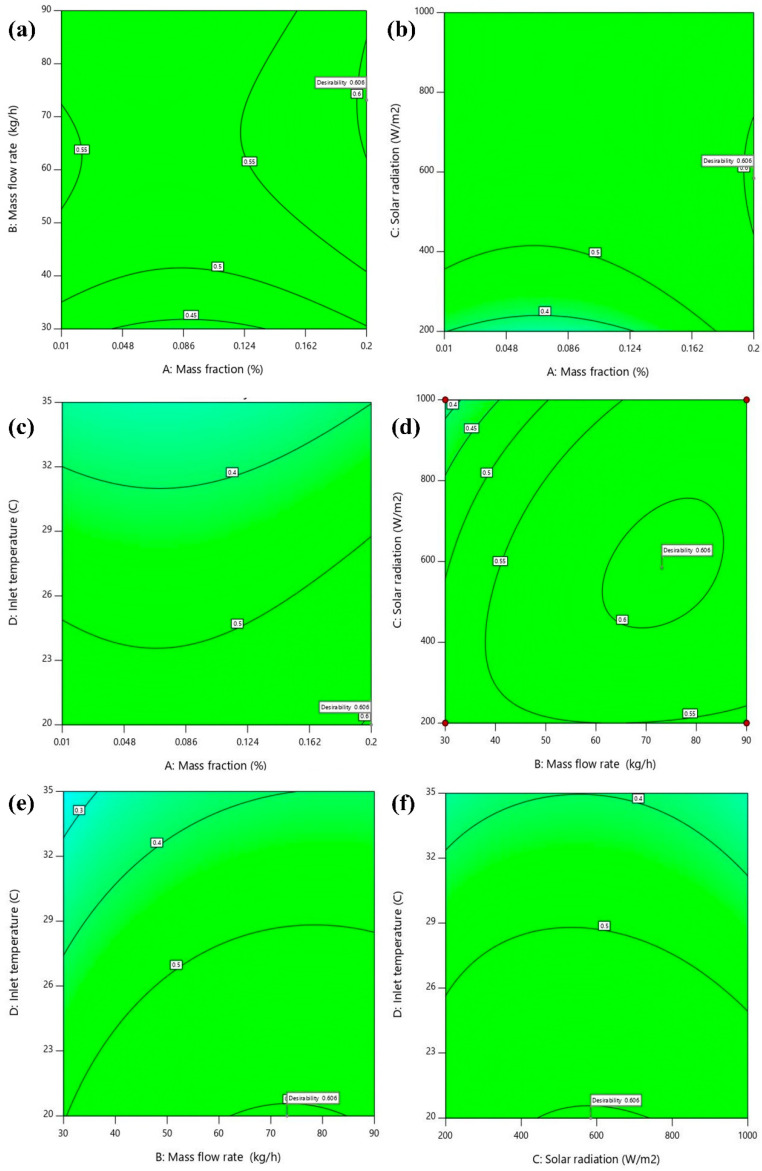
Variation in desirability graph based on the interaction of parameters (**a**) ${\dot{m}}_{nf}$ − $\varphi_{nf}$, (**b**) $\varphi_{nf}$ − I, (**c**) $\varphi_{nf}$ − T_i_, (**d**) ${\dot{m}}_{nf}$ − I, (**e**) ${\dot{m}}_{nf}$ − T_i_, and (**f**) I − T_i_.

**Table 1 nanomaterials-14-00774-t001:** PV/T components, dimensions and material properties used [33].

Component (in the Order from Top)	Dimension (m)	Thermal Conductivity (Wm^−1^K^−1^)	Specific Heat (Jkg^−1^K^−1^)	Density (Kgm^−3^)
Glass cover	1.64 × 0.2 × 0.0032	0.76	830	2200
EVA 1	1.64 × 0.2 × 0.0005	0.35	2090	960
PV Cell	1.64 × 0.2 × 0.0003	148	700	2330
EVA 2	1.64 × 0.2 × 0.0005	0.35	2090	960
Tedlar	1.64 × 0.2 × 0.0001	0.2	1250	1200
Absorber sheet	1.64 × 0.2 × 0.004	401	385	8960

**Table 2 nanomaterials-14-00774-t002:** Correlations used for thermophysical properties of a nanofluid.

References	Properties	Correlations
Pak and Cho [36]	Density	$\rho_{nf}=\varphi_{np}\rho_{np}+(1-\varphi_{np})\rho_{bf}$
Xuan and Roetzel [37]	Specific heat	$C_{p,\;\;nf}=\frac{\varphi_{np}\rho_{np}C_{p,np}+(1-\varphi_{np})\rho_{bf}}{\rho_{np}}$
Brinkman [38]	Viscosity	$\mu_{\;nf}=\frac{\mu_{bf}}{{\left(1-\varphi_{np}\right)}^{2.5}}$
Hamilton and Crosser [39]	Thermal conductivity	$K_{nf}=K_{bf}\frac{{(K}_{np}+2K_{bf})-2\varphi_{np}(K_{bf}-K_{np})}{{(K}_{np}+2K_{bf})+\varphi_{np}(K_{bf}-K_{np})}$

**Table 3 nanomaterials-14-00774-t003:** Parameters and values for energy and exergy calculations.

References	Parameters	Values
Khanjari et al. [34]	Reference temperature $\left(T_{ref}\right)$	298 K
Khanjari et al. [34]	Reference efficiency $\left(\eta_{ref}\right)$	18%
Kazemian et al. [40]	Cell temperature coefficient (β)	0.0045
Kazemian et al. [40]	Emissivity of glass (ℇ)	0.88
Eisapour et al. [41]	Sun temperature	5774 K
Present study	Aperture area	0.328 m^2^

**Table 4 nanomaterials-14-00774-t004:** Design parameter levels used in DoE.

Input Parameter		Levels of Input Factors		
	Representation	Low Level	Central Level	High Level
Nanofluid concentration	$\varphi_{nf}$ (wt%)	0.01	0.105	0.2
Nanofluid flow rate	${\dot{m}}_{nf}$ (kgh^−1^)	30	60	90
Solar radiation	I (Wm^−2^)	200	600	1000
Inlet temperature	T_i_ (°C)	20	27.5	35

**Table 5 nanomaterials-14-00774-t005:** Design matrix used for the optimization.

Sl. No.	Input Parameters				Output Response			
	φ (wt%)	$\dot{m}$ (kgh^−1^)	*I* (Wm^−2^)	*T_i_* (°C)	$\eta_{th}$ (%)	$\eta_{ele}$ (%)	$\eta_{ex,th}$ (%)	$\eta_{ex,ele}$ (%)
1	0.01	30	200	20	43.48	17.70	0.41	18.66
2	0.01	30	200	35	34.13	16.60	0.24	17.51
3	0.01	30	1000	20	50.79	14.76	2.66	15.56
4	0.01	30	1000	35	43.59	13.82	1.89	14.57
5	0.01	60	600	27.5	50.58	16.20	0.81	17.08
6	0.01	90	1000	35	55.74	14.83	1.06	15.64
7	0.01	90	1000	20	58.74	15.96	1.24	16.83
8	0.01	90	200	35	50.19	16.75	0.17	17.67
9	0.01	90	200	20	63.81	17.89	0.29	18.87
10	0.105	30	600	27.5	43.52	15.67	1.18	16.53
11	0.105	60	600	27.5	53.59	16.09	0.91	16.97
12	0.105	60	200	27.5	54.27	17.24	0.31	18.18
13	0.105	60	600	35	40.79	15.63	0.51	16.48
14	0.105	60	1000	27.5	54.25	15.03	1.53	15.85
15	0.105	60	600	27.5	53.59	16.09	0.91	16.97
16	0.105	60	600	27.5	53.59	16.09	0.91	16.97
17	0.105	60	600	20	56.57	16.65	1.03	17.56
18	0.105	90	600	27.5	48.96	16.33	0.51	17.22
19	0.2	30	1000	35	43.37	13.54	1.87	14.27
20	0.2	30	200	35	39.99	16.52	0.33	17.41
21	0.2	30	200	20	50.25	17.59	0.55	18.55
22	0.2	30	1000	20	50.12	14.47	2.59	15.25
23	0.2	60	600	27.5	57.41	15.97	1.04	16.84
24	0.2	90	1000	20	67.07	15.61	1.61	16.46
25	0.2	90	1000	35	63.50	14.50	1.37	15.29
26	0.2	90	200	35	57.30	16.70	0.23	17.60
27	0.2	90	200	20	72.91	17.82	0.39	18.79

**Table 6 nanomaterials-14-00774-t006:** ANOVA of PV/T thermal energy efficiency.

Source	Sum of Squares	Df	Mean Square	F Value	*p*-Value	Significance
Model	1856.36	14	132.60	11.86	<0.0001	significant
A-Mass fraction	143.72	1	143.72	12.86	0.0037	
B-Mass flow rate	1073.31	1	1073.31	96.01	<0.0001	
C-Solar radiation	24.11	1	24.11	2.16	0.1676	
D-Inlet temperature	402.43	1	402.43	36.00	<0.0001	
AB	26.42	1	26.42	2.36	0.1501	
AC	11.65	1	11.65	1.04	0.3274	
AD	0.5699	1	0.5699	0.0510	0.8252	
BC	22.96	1	22.96	2.05	0.1774	
BD	0.3069	1	0.3069	0.0275	0.8712	
CD	50.08	1	50.08	4.48	0.0559	
A^2^	31.27	1	31.27	2.80	0.1203	
B^2^	46.76	1	46.76	4.18	0.0634	
C^2^	36.21	1	36.21	3.24	0.0971	
D^2^	8.58	1	8.58	0.7675	0.3982	
Std. Dev.	3.34		R^2^	0.9326		
Mean	52.30		Adjusted R^2^	0.8540		
C.V. %	6.39		Predicted R^2^	0.7051		
PRESS	692.87		Adeq Precision	15.3254		

**Table 7 nanomaterials-14-00774-t007:** ANOVA of PV/T electrical energy efficiency.

Source	Sum of Squares	Df	Mean Square	F Value	*p*-Value	Significance
Model	35.74	14	2.55	559.26	<0.0001	significant
A-Mass fraction	0.1824	1	0.1824	39.96	<0.0001	
B-Mass flow rate	1.83	1	1.83	401.63	<0.0001	
C-Solar radiation	27.61	1	27.61	6048.10	<0.0001	
D-Inlet temperature	5.08	1	5.08	1112.97	<0.0001	
AB	0.0001	1	0.0001	0.0241	0.8793	
AC	0.0528	1	0.0528	11.57	0.0053	
AD	0.0003	1	0.0003	0.0684	0.7981	
BC	0.7936	1	0.7936	173.85	<0.0001	
BD	0.0133	1	0.0133	2.92	0.1135	
CD	0.0068	1	0.0068	1.48	0.2473	
A^2^	0.0064	1	0.0064	1.41	0.2582	
B^2^	0.0464	1	0.0464	10.16	0.0078	
C^2^	1.666 × 10^−7^	1	1.666 × 10^−7^	0.0000	0.9953	
D^2^	0.0001	1	0.0001	0.0182	0.8950	
Std. Dev.	0.0676		R^2^	0.9985		
Mean	16.00		Adjusted R^2^	0.9967		
C.V. %	0.4221		Predicted R^2^	0.9925		
PRESS	0.2698		Adeq Precision	86.9582		

**Table 8 nanomaterials-14-00774-t008:** ANOVA of PV/T thermal exergy efficiency.

Source	Sum of Squares	Df	Mean Square	F Value	*p*-Value	Significance
Model	12.27	14	0.8762	56.21	<0.0001	significant
A-Mass fraction	0.0796	1	0.0796	5.11	0.0432	
B-Mass flow rate	1.31	1	1.31	84.09	<0.0001	
C-Solar radiation	9.22	1	9.22	591.71	<0.0001	
D-Inlet temperature	0.5330	1	0.5330	34.19	<0.0001	
AB	0.0293	1	0.0293	1.88	0.1951	
AC	0.0031	1	0.0031	0.1994	0.6631	
AD	0.0005	1	0.0005	0.0351	0.8544	
BC	0.6766	1	0.6766	43.41	<0.0001	
BD	0.0898	1	0.0898	5.76	0.0335	
CD	0.0957	1	0.0957	6.14	0.0291	
A^2^	0.0311	1	0.0311	1.99	0.1834	
B^2^	0.0024	1	0.0024	0.1547	0.7010	
C^2^	0.0291	1	0.0291	1.87	0.1966	
D^2^	0.0040	1	0.0040	0.2552	0.6226	
Std. Dev.	0.1249		R^2^	0.9850		
Mean	0.9852		Adjusted R^2^	0.9675		
C.V. %	12.67		Predicted R^2^	0.8994		
PRESS	1.25		Adeq Precision	26.3117		

**Table 9 nanomaterials-14-00774-t009:** ANOVA of PV/T electrical exergy efficiency.

Source	Sum of Squares	Df	Mean Square	F Value	*p*-Value	Significance
Model	39.74	14	2.84	559.26	<0.0001	significant
A-Mass fraction	0.2028	1	0.2028	39.96	<0.0001	
B-Mass flow rate	2.04	1	2.04	401.63	<0.0001	
C-Solar radiation	30.70	1	30.70	6048.10	<0.0001	
D-Inlet temperature	5.65	1	5.65	1112.97	<0.0001	
AB	0.0001	1	0.0001	0.0241	0.8793	
AC	0.0587	1	0.0587	11.57	0.0053	
AD	0.0003	1	0.0003	0.0684	0.7981	
BC	0.8823	1	0.8823	173.85	<0.0001	
BD	0.0148	1	0.0148	2.92	0.1135	
CD	0.0075	1	0.0075	1.48	0.2473	
A^2^	0.0072	1	0.0072	1.41	0.2582	
B^2^	0.0516	1	0.0516	10.16	0.0078	
C^2^	1.852 × 10^−7^	1	1.852 × 10^−7^	0.0000	0.9953	
D^2^	0.0001	1	0.0001	0.0182	0.8950	
Std. Dev.	0.0712		R^2^	0.9985		
Mean	16.88		Adjusted R^2^	0.9967		
C.V. %	0.4221		Predicted R^2^	0.9925		
PRESS	0.300		Adeq Precision	86.9582		

**Table 10 nanomaterials-14-00774-t010:** Coefficients of variables and corresponding *p*-value for each model.

Sl. No.	Output Responses and *p*-Values							
	Thermal Energy Efficiency $\left(\eta_{th}\right)$		Electrical Energy Efficiency $\left(\eta_{ele}\right)$		Thermal Exergy Efficiency $\left(\eta_{ex,th}\right)$		Electrical Exergy Efficiency $\left(\eta_{ex,ele}\right)$	
	Value	*p*-Value	Value	*p*-Value	Value	*p*-Value	Value	*p*-Value
$a_{0}$	51.54	-	16.12	-	0.85	-	17.00	-
$a_{1}$	2.83	0.0037	−0.10	39.96	0.067	0.043	−0.11	<0.0001
$a_{2}$	7.72	<0.0001	0.32	401.63	−0.27	<0.0001	0.34	<0.0001
$a_{3}$	1.16	0.17	−1.24	6048.10	0.72	<0.0001	−1.31	<0.0001
$a_{4}$	−4.73	<0.0001	−0.53	1112.97	−0.17	<0.0001	−0.56	<0.0001
$a_{5}$	1.29	0.15	−0.0026	0.024	0.042	0.19	−0.0028	0.88
$a_{6}$	−0.85	0.33	−0.057	11.57	0.014	0.66	−0.061	0.0053
$a_{7}$	−0.19	0.83	0.0044	0.068	−0.0059	0.85	0.0047	0.79
$a_{8}$	−1.20	0.18	0.22	173.85	−0.21	<0.0001	0.24	<0.0001
$a_{9}$	−0.14	0.87	−0.029	2.92	0.075	0.034	−0.030	0.11
$a_{10}$	1.77	0.056	0.021	1.48	−0.077	0.029	0.022	0.25
$a_{11}$	3.49	0.120	−0.05	1.41	0.11	0.18	−0.053	0.26
$a_{12}$	−4.26	0.063	−0.13	10.16	0.031	0.70	−0.14	0.0078
$a_{13}$	3.75	0.097	−0.0003	0.0000	0.11	0.19	−0.0003	0.99
$a_{14}$	−1.83	0.39	0.0057	0.018	−0.039	0.62	0.0060	0.89

**Table 11 nanomaterials-14-00774-t011:** Optimized values of input parameters from optimization.

	Optimized Input Parameters and Response Values								
	Optimized Input Parameters				Predicted Response Values				Desirability
Solutions	φ (wt%)	$\dot{m}$ (kgh^−1^)	*I* (Wm^−2^)	*T_i_* (°C)	$\eta_{th}$ (%)	$\eta_{ele}$ (%)	$\eta_{ex,th}$ (%)	$\eta_{ex,ele}$ (%)	
1	0.200	71.848	581.214	20.000	65.034	16.679	1.013	17.587	0.606
2	0.165	68.287	615.853	20.000	61.180	16.611	1.014	17.515	0.584
3	0.010	69.517	829.766	20.000	58.633	16.164	1.342	17.044	0.572

## Data Availability

All data is available in the manuscript.

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
