# Peer review of "Performance Prediction and Optimization of Nanofluid-Based PV/T Using Numerical Simulation and Response Surface Methodology"

_nanomaterials, 2024, doi:10.3390/nano14090774_

Round 1

Reviewer 1 Report

Comments and Suggestions for Authors

The present work performed the prediction and optimization of MXene/water nanofluid based PV/T by numerical simulation and response surface methodology. The results showed that electrical efficiency, thermal exergy efficiency, and electrical exergy efficiency are greatly influenced by incident solar radiation. The work could shed some light in application of nanofluid. However, there are several issues should be addressed before further consideration of this paper.

1. There are so many types of nanofluid, why the MXene/water nanofluid is considered here?

2. The details of thermophysical properties of nanofluid should be given. Not just the correlations.

3. The nanofluid flow was assumed as laminar, why?

Comments on the Quality of English Language

The present work performed the prediction and optimization of MXene/water nanofluid based PV/T by numerical simulation and response surface methodology. The results showed that electrical efficiency, thermal exergy efficiency, and electrical exergy efficiency are greatly influenced by incident solar radiation. The work could shed some light in application of nanofluid. However, there are several issues should be addressed before further consideration of this paper.

1. There are so many types of nanofluid, why the MXene/water nanofluid is considered here?

2. The details of thermophysical properties of nanofluid should be given. Not just the correlations.

3. The nanofluid flow was assumed as laminar, why?

Reviewer 2 Report

Comments and Suggestions for Authors

In this paper, the authors have a numerical simulation of MXene/water nanofluid-based PV/T was con- ducted in ANSYS Fluent®. The interaction of different operating parameters (nanofluid mass fraction, mass flow rate, inlet temperature and incident radiation) on the output response of the system (thermal efficiency, electrical efficiency, thermal exergy efficiency, and electrical exergy efficiency) was 13 studied using a predictive model generated using response surface methodology (RSM). The analysis of variance (ANOVA) method was used to evaluate the significance of input parameters affecting the energy and exergy efficiencies of the nanofluid-based PV/T system.

 In my opinion, the work is new, and the manuscript can be considered for publication in the journal of nanomaterials. Before then, it must undergo the following revision according to the following comments:

 The current abstract must be improved and try to make it short and must contain answers to the following questions: Why is this particular computational study is important?

  1. What are the important results of this study that are different from others?
  2. Give more arguments in favour of the numerical methods studied in the literature.
  3. The major contributions of this paper must be strongly highlighted in the concluding part of the introduction.
  4. In the present version of the model, it seems that the different diffusions play an important role in bisecting the domain. Is there a limit to the magnitude of both diffusion coefficients?
  5. The authors have also used the finite volume method. Can you give a short reference to where your scheme for the finite volume came from?
  6. I also think it will be nice if authors also give details on the order of convergence for the scheme somewhere in the text.
  7. Provide detailed future research directions in the conclusion.
  8. I recommend that the entire manuscript be carefully and thoroughly checked for typos and also improved to attract a wider readership.
  9. Elobrate the application of flow model and methods?
  10. Improve the introduction section with recent publication as: (1) Chemically reactive aspects of stagnation-point boundary layer flow of second-grade nanofluid over an exponentially stretching surface; (2) Bejawada, S.G., Nandeppanavar, M.M. Effect of thermal radiation on magnetohydrodynamics heat transfer micropolar fluid flow over a vertical moving porous plate. Exp. Comput. Multiph. Flow 5, 149–158 (2023). https://doi.org/10.1007/s42757-021-0131-5
Comments on the Quality of English Language

It should be improved by editor.

Reviewer 3 Report

Comments and Suggestions for Authors

The authors numerically simulated the MXene/ water nanofluid-based PV/T, using the response surface methodology (RSM) to generate a predictive model and using the analysis of variance (ANOVA) method to evaluate the influence of input parameters on the energy and exergy efficiency of the system. In my opinion, the following issues should be clarified before the article reaches the level required for publication in Nanomaterials. My main comments are summarized as follows:

1.      The format of serial numbers introduced in the literature and formula in this paper needs to be adjusted (such as " Error! Reference source not found ").

2.      Regarding the simulation of fluid-particle flow, the author needs to examine more published papers. Such as https://doi.org/10.1007/s11630-024-1860-0https://doi.org/10.1016/j.ijheatmasstransfer.2013.06.005; https://doi.org/10.1016/j.ijheatmasstransfer.2017.10.130.

3.      The figure and table of this paper still need to be adjusted. Part of the contents of Fig. 9 ~ Fig. 12 were obscured. Number the subgraphs in Fig.14. The text in Table 3 must be in the same format.

4.      In the Abstract, "incident radiation (581.214)" lacks units.

5.      Fig. 2 should be located below section 2.1, and "Methodology" instead of "Geometric Model" should be used in Fig.1. It is suggested to change "2.1 Geometric Model" to "2.1 Methodology and Geometric Model" and place Fig.2 under Table 1.

6.      Section 3.5 mentions, "The thermal energy and exergy efficiency of the PV/T has the highest dependence on mass flow rate." Still, as can be seen from Fig. 13, the thermal exergy efficiency has the highest dependence on solar radiation. The two are contradictory. Please correct.

7.      It can be seen from the fourth figure in Fig.14 that 0.606 is the maximum value of Desirability. Why is the maximum value of Desirability 0.611 in Table 11? Please explain.

8.      It can be seen from Figures 3, 5, 6, and 14 that Desirability increases when inlet temperature (Ti) is reduced. Why does this paper not use a lower inlet temperature than 20℃ for the experiment?

9.      I suggest that the author redraw all data to make them more standardized and understandable.

Comments on the Quality of English Language

Suggest the author to express the relevant content in more understandable language.

Reviewer 4 Report

Comments and Suggestions for Authors

I reallz had a hard time to read this manucript since there is an Error! Reference source not found in all the place should indicate the figure even I try very hard to understand what author is tring to describe.   Also, a lot of reference is missing. i.e. in line 82 The literature review shows that MXene nanoparticle dispersions signifi cantly enhanced the thermal properties of the base fluid.  additionally, the author provide compartion of real data with simulation. However, I can not find the experiment part or the data source. I am really sorry I can not further evaluate this manuscript with this condition and have to suggest to reject it.

Comments on the Quality of English Language

The Quality of english writing is fair.

Round 2

Reviewer 1 Report

Comments and Suggestions for Authors

accept

Comments on the Quality of English Language

accept

Author Response

The authors thank the reviewer for conducting a professional, exceptional, and timely review process. The authors would like to believe that the responses to the comments were satisfactory and led to the positive decision made by the reviewer.

Reviewer 2 Report

Comments and Suggestions for Authors

It can be accepted.

Comments on the Quality of English Language

It can be accepted.

Author Response

The authors thank the Reviewer for conducting a professional, exceptional, and timely review process. The authors would like to believe that the responses to the comments were satisfactory and led to the positive decision made by the reviewer. The English language has been modified in the manuscript as per the Reviewer's suggestion.

Reviewer 3 Report

Comments and Suggestions for Authors

The Authors answered all my queries and those of another Referee. I cannot ask for too much more. However, this revised manuscript still has several areas where the writing is not standardized, and some of the figures need further clarity and improvement.

Comments on the Quality of English Language

I have no comments on the quality of the English language.

Author Response

The authors thank the Reviewer for the compliments. The authors acknowledge the dedication of the Reviewer in completing a professional, timely and efficient review process. Modifications were made in the manuscript to improve the quality of the paper. The quality of a few images was improved.

Reviewer 4 Report

Comments and Suggestions for Authors

I understand the situation of the previous version. Now I can finally read the manuscript.  

In this manuscipt the author suggest to use numarical simulation method combine to optimize the fluid PV/T system. 

The method seems a extensive method for all kind of fluid instead of specify for nanofluid or Mxene/water fluid. The Mxene is only for validation without consider the material science. I did not get why the author want to specify it. 

I would recomend the manuscipt go to other journal since this research did not related to insight of material science.

 For the Validating in Figure 5. author provide numerical comparesion between the present study with literatures. However, it is not possible to validate method without static comparsion although the trend looks simillar.

 For the previous comment of the missing citation(The literature review shows that MXene nanoparticle dispersions significantly en- 94 hanced the thermal properties of the base fluid.), after carefully read the introduction, I still can not get the same conclusion that author made. I would recommend the author the explain the logic.  Also this sentence happen in the new paragraph would make reader hard to know it is the conclusion is from which part of the manuscript. 

I ask the author understanding the grammarly correct has nothing related to the review quality. Also I ask author dedicate more time to understand the review comment.  The auther provide compare between predict and actual data(see in figure 6, 7, 8, 9)  It is not true the research is entirely based on numerical simulation if there is a real data compared. As the previous comment, there is no clear indicate for the source of the data and I was confused that the data can be possible source from experiment did by the author.  And I can not find the data source in the manuscript still.

A lot of component in the figure is not explain in the figure caption . For example, in  Figure 10, 11, 12, 13 waht is the red point means in the 3d plot.

Also the quality of citation should be improve. Citing a reference with scientisfic publication  is prefered compare to an online learning website which might have chance to disappear at some moment.  

Also I suggest the author to check the significant figures. 

minor suggestion,

Please check the superscript and subscript.  for example in table 1.((Kg/m3) should be (Kg/m3 ) line 210(The number of elements of the domain varied from 1.5 to 5 × 105should be 105 .)

Author Response

Please find the response to the comments in the file attached.
